# Effectiveness of ChAdOx1-S COVID-19 booster vaccination against the Omicron and Delta variants in England

Freja Cordelia Møller Kirsebom [1] ✉, Nick Andrews[1,2], Ruchira Sachdeva[1], Julia Stowe[1], Mary Ramsay[1,2] & Jamie Lopez Bernal[1,2,3] ✉

Despite the availability of the ChAdOx1-S booster vaccine, little is known about the real-world effectiveness although clinical trials have demonstrated enhanced immunity following a ChAdOx1-S booster. In England 43,171 individuals received a ChAdOx1-S booster whilst 13,038,908 individuals received BNT162b2 in the same period. ChAdOx1-S booster recipients were more likely to be female (adjusted odds ratio (OR) 1.67 (1.64-1.71)), in a clinical risk group (adjusted OR 1.58 (1.54-1.63)), in the clinically extremely vulnerable group (adjusted OR 1.84 (1.79-1.89)) or severely immunosuppressed (adjusted OR 2.05 (1.96-2.13)). The effectiveness of the ChAdOx1-S and BNT162b2 boosters is estimated here using a test-negative case-control study. Protection against symptomatic disease with the Omicron variant peaks at 66.1% (16.6 to 86.3%) and 68.5% (65.7 to 71.2%) for the ChAdOx1-S and BNT162b2 boosters in older adults. Protection against hospitalisation peaks at 82.3% (64.2 to 91.3%) and 90.9% (88.7 to 92.7%). For Delta, effectiveness against hospitalisation is 80.9% (15.6% to 95.7%) and 93.9% (92.8% to 94.9%) after ChAdOx1-S and BNT162b2 booster vaccination. This study supports the consideration of ChAdOx1-S booster vaccination for protection against severe COVID-19 in settings yet to offer boosters and suggests that individuals who received a ChAdOx1-S booster do not require re-vaccination ahead of others.

Real-world studies on the vaccine effectiveness (VE) and duration of protection conferred by booster vaccines against mild and severe outcomes of SARS-CoV-2 infection have primarily focussed on mRNA vaccine boosters[1–4]. Over 1.6 billion booster doses have been administered globally, including with the adenoviral vector-based ChAdOx1-S (Vaxzevria/Covishield, AstraZeneca) vaccine although it is unclear how many ChAdOx1-S boosters have been administered[5]. The ChAdOx1-S vaccine is in use as a booster dose as part of both homologous and heterologous vaccine programmes in South America, Africa, and Asia[6–8], and has been relied upon particularly in low- and middle-income countries due to the lower cost and the relative

logistical ease of use as the cold-chain requirements are less extensive than those of the mRNA vaccines.

Real-world VE studies have found one and two doses of the ChAdOx1-S vaccine to be moderately effective against mild disease and highly effective against severe COVID-19, with VE against symptomatic disease peaking around 50% whilst VE against hospitalisation peaks at 85–90%[9–12]. Nonetheless, waning over time occurs after a primary course (two doses)[13]. Waning of protection against symptomatic disease caused by the Omicron variant occurs from a month after the second dose. Protection against more severe outcomes (hospitalisation and death) is maintained for longer, but by 25 or more weeks after

[1]UK Health Security Agency, London, UK. [2]NIHR Health Protection Research Unit in Vaccines and Immunisation, London School of Hygiene and Tropical Medicine, London, UK. [3]NIHR Health Protection Research Unit in Respiratory Infections, Imperial College London, London, UK.
✉e-mail: Freja.kirsebom@ukhsa.gov.uk; Jamie.LopezBernal2@ukhsa.gov.uk

a second dose protection against hospitalisation with Omicron is estimated at around 30%[14]. Clinical trial data have indicated that ChAdOx1-S induces high titres of anti-spike IgG with good neutralising capacity as well as a strong cellular immune response when given as a booster following primary vaccination[15–17], although to date there are limited data on the real-world effectiveness. Given the potential for widespread use of ChAdOx1-S vaccine as a booster, there is an urgent need to understand the level and duration of protection conferred in real-world settings.

The UK COVID-19 vaccination programme has been in place since December 2020 with primary courses of two doses of either BNT162b2 (Pfizer-BioNTech, Comirnaty®), ChAdOx1-S or mRNA-1273 (Spikevax, Moderna). From May 2021, ChAdOx1-S primary immunisation was not recommended as first-line vaccination for individuals not in a clinical risk group aged under 40 years following reports of rare cases of concurrent thrombosis and thrombocytopenia after the first ChAdOx1-S dose[18,19]. Booster vaccination with either BNT162b2 or a half dose (50 μg) of mRNA-1273 was introduced in September 2021 to adults over 50 years old and those in risk groups, and in November 2021 expanded to all adults. A small number of individuals who had received at least one dose of ChAdOx1-S previously and for whom vaccination with both BNT162b2 and mRNA-1273 were clinically contraindicated, were recommended booster vaccination with the ChAdOx1-S vaccine[20]. Clinical contraindications included individuals with a prior allergic reaction to any component of the vaccine, e.g. polyethylene glycol, or individuals who had a previous systemic anaphylaxis reaction to a COVID-19 vaccine[20]. ChAdOx1-S was in some cases also offered for logistical reasons, for example, to housebound individuals.

In this study, we estimate VE against symptomatic disease and hospitalisation following Delta or Omicron infection after booster vaccination in adults in England who received a ChAdOx1-S booster vaccine as compared to those who received a BNT162b2 booster following a ChAdOx1-S primary course.

## Results

### Characteristics of individuals who received a ChAdOx1-S booster
To investigate the demographic characteristics of individuals who received a ChAdOx1-S booster vaccination, data were extracted from the NIMS on 14 March 2022 on all adults aged 18 years and older in England who had received a ChAdOx1-S primary course followed by a ChAdOx1-S or BNT162b2 booster (Table 1). In total, there were 43,171 individuals who received a ChAdOx1-S booster dose as compared to 13,038,908 individuals who received BNT162b2. Those who received a ChAdOx1-S booster were more likely to be female (adjusted odds ratio (OR) 1.67 (1.64–1.71)), from London (adjusted OR 2.84 (2.73–2.95), health and social care workers (adjusted OR 1.58 (1.54–1.63)), in a clinical risk group (adjusted OR 1.58 (1.54–1.63)), in the clinically extremely vulnerable (CEV) group (adjusted OR 1.84 (1.79–1.89)) or in the severely immunosuppressed group (adjusted OR 2.05 (1.96–2.13)) (Table 1).

### Effectiveness of the ChAdOx1-S booster against mild disease
Between 29 November 2021 and 17 February 2022 (the Omicron variant study period), there was a total of 457,377 negative and 434,514 positive eligible tests while between 13 September 2021 and 9 January 2022 (the Delta variant study period) there was a total of 469,976 negative and 172,223 positive eligible tests. Eligible tests included those from individuals aged 40 years and older, with a test date within 10 days of the symptom onset date which could be linked to the National Immunisation Management system (full details of data sources in Table 2). A description of eligible tests from symptomatic cases is included in Supplementary Table 3. The Omicron sub-lineages BA.1 and BA.2 were both circulating during this period. Previously we have

not found a difference in VE between BA.1 and BA.2[21]. Therefore, we combined BA.1 and BA.2 cases for the Omicron analyses.

Following a ChAdOx1-S primary vaccination course, VE against symptomatic disease caused by the Omicron variant was 8.0% (6.0–9.9%) and 19.5% (11.7–26.6%) for those aged 40–64 years and those aged 65 years and older, respectively, after 25 or more weeks (Table 3). In those aged 40–64 years, VE against symptomatic disease increased to 61.2% (40.9–74.6%) one week after receiving a ChAdOx1-S booster as compared to 58.2% (57–59.4%) amongst those who received a BNT162b2 booster. The protection of the ChAdOx1-S booster waned to 37.2% (−44.1 to 72.6%) at 15 or more weeks after receiving the booster, as compared to 30.6% (26.8–34.3%) over the same period amongst those who received a BNT162b2 booster.

Similar levels of protection against symptomatic disease were observed in those aged 65 years and older (Table 3). After receiving the booster, VE peaked at 66.1% (16.6–86.3%) and 68.5% (65.7–71.2%) amongst those who received the ChAdOx1-S and BNT162b2 booster vaccines, respectively. In older adults, this protection waned to 44.5% (22.4–60.2%) and 54.1% (50.5–57.5%) after 5–9 weeks amongst those who received the ChAdOx1-S and BNT162b2 booster vaccines, respectively. Beyond 10 weeks the analysis was limited by small numbers but there was evidence of further waning.

For those aged 40–64 years, VE against symptomatic disease following infection with the Delta variant was 85.4% (77.4–90.6%) and 91.9% (91.6–92.1%) for the ChAdOx1-S and BNT162b2 booster vaccines, respectively, in any period from 7 days post booster vaccination (Supplementary Table 4). For those aged 65 years and older VE were 56.3% (30.0–72.7%) and 90.2% (89.4–90.9%) for the ChAdOx1-S and BNT162b2 booster vaccines, respectively. There were not enough cases to estimate VE in more granular intervals post the booster dose.

### Effectiveness of the ChAdOx1-S booster against hospitalisation
The effectiveness of the ChAdOx1-S booster against hospitalisation was estimated following infection with either Delta or Omicron variants, by linkage of Pillar 1 and Pillar 2 tests to the secondary care (SUS) data (Table 4, Fig. 1) and by linkage of Pillar 2 tests to accident and emergency (ECDS) data (Supplementary Tables 3 and 5).

Following linkage to the SUS data, there were a total of 17,377 negative and 5850 positive (Delta) eligible tests between 13 September 2021 and 9 January 2022, while there were a total of 9021 negative and 2031 positive (Omicron) eligible tests between 29 November 2021 and 2 February 2022. Following linkage to the ECDS data, there was a total of 469,976 negative and 1352 positive (Delta) eligible tests between 13 September 2021 and 9 January 2022, while there was a total of 457,377 negative and 509 positive (Omicron) eligible tests between 29 November 2021 and 17 February 2022. A description of eligible tests from all hospitalised cases is included in Supplementary Tables 3 and 6.

Amongst those aged 65 years and older, protection against hospitalisation (defined as requiring a stay of two or more days with severe respiratory disease) following Omicron infection was 61.0% (49.8–69.7%) at 25 weeks after a second dose (Table 4, Fig. 1). This increased to 82.3% (64.2–91.3%) one or more weeks after receiving a ChAdOx1-S booster, as compared to 90.9% (88.7–92.7%) for those who received a BNT162b2 booster. Protection against hospitalisation following Delta infection was also enhanced by the ChAdOx1-S booster in older adults. VE against Delta hospitalisation was 73.4% (70.4–76.2%) at 25 weeks after receiving a second dose. This increased to 80.9% (15.6–95.7%) and 93.9% (92.8–94.9%), respectively, one week after vaccination with the ChAdOx1-S and BNT162b2 booster vaccines, respectively (Table 4, Fig. 1). VE estimates derived using a broader definition of hospitalisation with the ECDS-linked data after a booster dose of ChAdOx1-S were substantially lower at around 65% with either variant (Supplementary Table 5).

**Table 1 | Demographic characteristics of the adults who received a ChAdOx1-S booster as compared to a BNT162b2 booster, following a ChAdOx1-S primary course, in England**

| | | Booster manufacturer | | | | Logistic regression outputs | | | |
| --- | --- | --- | --- | --- | --- | --- | --- | --- | --- |
| | | ChAdOx1-S | | BNT162b2 | | Univariable | | Multivariable | |
| | | Count | % | Count | % | OR | CI (95%) | OR | CI (95%) |
| Total | | 43,171 | 0.3 | 13,038,908 | 99.7 | | | | |
| Sex | Female | 28,234 | 65.4 | 6,755,497 | 51.8 | 1.78 | (0.52–1.78) | 1.67 | (1.64–1.71) |
| | Male | 14,755 | 34.2 | 6,276,151 | 48.1 | Baseline | | Baseline | |
| | Missing | 182 | 0.4 | 7260 | 0.1 | 10.66 | (0–10.66) | 12.09 | (10.17–14.38) |
| Age | 18–19 years | 108 | 0.3 | 42,563 | 0.3 | 0.87 | (0.72–1.05) | 0.85 | (0.7–1.03) |
| | 20–24 years | 596 | 1.4 | 191,295 | 1.5 | 1.07 | (0.98-1.16) | 0.93 | (0.85–1.02) |
| | 25–29 years | 829 | 1.9 | 251,910 | 1.9 | 1.13 | (1.05–1.21) | 0.90 | (0.84–0.98) |
| | 30–34 years | 1117 | 2.6 | 357,025 | 2.7 | 1.07 | (1-1.14) | 0.87 | (0.81–0.93) |
| | 35–39 years | 1439 | 3.3 | 456,238 | 3.5 | 1.08 | (1.02–1.15) | 0.91 | (0.86–0.97) |
| | 40–44 years | 2534 | 5.9 | 1,100,893 | 8.4 | 0.79 | (0.75–0.83) | 0.79 | (0.75–0.83) |
| | 45–49 years | 3076 | 7.1 | 1,286,227 | 9.9 | 0.82 | (0.78–0.86) | 0.83 | (0.79–0.87) |
| | 50–54 years | 4324 | 10.0 | 1,653,865 | 12.7 | 0.90 | (0.86–0.93) | 0.91 | (0.87–0.94) |
| | 55–59 years | 4840 | 11.2 | 1,668,410 | 12.8 | 0.99 | (0.95–1.04) | 0.99 | (0.95–1.04) |
| | 60–64 years | 4309 | 10.0 | 1,476,385 | 11.3 | Baseline | | Baseline | |
| | 65–69 years | 4129 | 9.6 | 1,378,031 | 10.6 | 1.03 | (0.98–1.07) | 1.04 | (0.99–1.08) |
| | 70–74 years | 4657 | 10.8 | 1,495,452 | 11.5 | 1.07 | (1.02–1.11) | 1.10 | (1.06–1.15) |
| | 75–79 years | 3705 | 8.6 | 971,901 | 7.5 | 1.31 | (1.25–1.36) | 1.30 | (1.25–1.36) |
| | 80–84 years | 2850 | 6.6 | 344,575 | 2.6 | 2.83 | (2.7–2.97) | 2.44 | (2.32–2.56) |
| | 85–89 years | 2351 | 5.4 | 204,320 | 1.6 | 3.94 | (3.75–4.15) | 2.91 | (2.76–3.06) |
| | 90 and over | 2307 | 5.3 | 159,818 | 1.2 | 4.95 | (4.7–5.2) | 3.16 | (3–3.33) |
| Ethnicity | African | 640 | 1.5 | 130,728 | 1.0 | 1.53 | (1.41–1.65) | 0.79 | (0.73–0.85) |
| | Any other Asian background | 518 | 1.2 | 168,354 | 1.3 | 0.96 | (0.88–1.05) | 0.66 | (0.61–0.73) |
| | Any other Black background | 301 | 0.7 | 44,674 | 0.3 | 2.11 | (1.88–2.36) | 1.20 | (1.07–1.35) |
| | Any other White background | 667 | 1.5 | 167,005 | 1.3 | 1.10 | (1.06–1.14) | 0.93 | (0.89–0.97) |
| | Any other ethnic group | 186 | 0.4 | 50,080 | 0.4 | 1.25 | (1.16–1.35) | 0.86 | (0.8–0.93) |
| | Any other mixed background | 2648 | 6.1 | 753,121 | 5.8 | 1.16 | (1–1.34) | 0.88 | (0.76–1.02) |
| | Bangladeshi or British Bangladeshi | 228 | 0.5 | 70,704 | 0.5 | 1.01 | (0.88–1.15) | 0.59 | (0.51–0.67) |
| | British, Mixed British | 31,184 | 72.2 | 9,744,021 | 74.7 | Baseline | | Baseline | |
| | Caribbean | 748 | 1.7 | 66,580 | 0.5 | 3.51 | (3.26–3.78) | 1.51 | (1.4–1.63) |
| | Chinese | 171 | 0.4 | 61,295 | 0.5 | 0.87 | (0.75–1.01) | 0.75 | (0.64–0.87) |
| | Indian or British Indian | 969 | 2.2 | 337,218 | 2.6 | 0.90 | (0.84–0.96) | 0.69 | (0.64–0.73) |
| | Irish | 423 | 1.0 | 86,077 | 0.7 | 1.54 | (1.39–1.69) | 1.02 | (0.93–1.13) |
| | Pakistani or British Pakistani | 478 | 1.1 | 150,997 | 1.2 | 0.99 | (0.9–1.08) | 0.81 | (0.74–0.89) |
| | White and Asian | 72 | 0.2 | 24,061 | 0.2 | 0.94 | (0.74–1.18) | 0.82 | (0.65-1.04) |
| | White and Black African | 61 | 0.1 | 18,295 | 0.1 | 1.04 | (0.81–1.34) | 0.75 | (0.58–0.96) |
| | White and Black Caribbean | 132 | 0.3 | 21,445 | 0.2 | 1.92 | (1.62–2.28) | 1.31 | (1.1–1.56) |
| | Missing | 3745 | 8.7 | 1,144,253 | 8.8 | 1.02 | (0.99–1.06) | 1.08 | (1.05–1.12) |
| Region | East of England | 3999 | 9.3 | 1,451,568 | 11.1 | Baseline | | Baseline | |
| | London | 10,412 | 24.1 | 1,505,628 | 11.5 | 2.51 | (2.42–2.6) | 2.84 | (2.73–2.95) |
| | Midlands | 6116 | 14.2 | 2,565,254 | 19.7 | 0.87 | (0.83–0.9) | 0.88 | (0.85–0.92) |
| | North East and Yorkshire | 4768 | 11.0 | 1,908,956 | 14.6 | 0.91 | (0.87–0.95) | 0.86 | (0.82–0.89) |
| | North West | 5871 | 13.6 | 1,638,879 | 12.6 | 1.30 | (1.25–1.35) | 1.32 | (1.26–1.37) |
| | South East | 7110 | 16.5 | 2,500,581 | 19.2 | 1.03 | (0.99–1.07) | 1.13 | (1.08–1.17) |
| | South West | 4576 | 10.6 | 1,433,381 | 11.0 | 1.16 | (1.11–1.21) | 1.16 | (1.11–1.21) |
| | Missing | 319 | 0.7 | 34,661 | 0.3 | 3.34 | (2.98–3.75) | | |
| IMD[a] | 1 | 3418 | 7.9 | 977,959 | 7.5 | Baseline | | Baseline | |
| | 2 | 4318 | 10.0 | 1,051,604 | 8.1 | 1.17 | (1.12–1.23) | 1.00 | (0.95–1.04) |
| | 3 | 4492 | 10.4 | 1,132,655 | 8.7 | 1.13 | (1.09–1.19) | 0.95 | (0.91–0.99) |
| | 4 | 4457 | 10.3 | 1,238,022 | 9.5 | 1.03 | (0.99–1.08) | 0.92 | (0.88–0.96) |
| | 5 | 4408 | 10.2 | 1,314,814 | 10.1 | 0.96 | (0.92–1) | 0.89 | (0.85–0.93) |
| | 6 | 4550 | 10.5 | 1,391,251 | 10.7 | 0.94 | (0.9–0.98) | 0.89 | (0.85–0.93) |
| | 7 | 4367 | 10.1 | 1,420,830 | 10.9 | 0.88 | (0.84–0.92) | 0.86 | (0.82–0.9) |

**Table 1 (continued) | Demographic characteristics of the adults who received a ChAdOx1-S booster as compared to a BNT162b2 booster, following a ChAdOx1-S primary course, in England**

| | | Booster manufacturer | | | | Logistic regression outputs | | | |
|---|---|---|---|---|---|---|---|---|---|
| | | ChAdOx1-S | | BNT162b2 | | Univariable | | Multivariable | |
| | | Count | % | Count | % | OR | CI (95%) | OR | CI (95%) |
| | 8 | 4492 | 10.4 | 1,466,406 | 11.2 | 0.88 | (0.84–0.92) | 0.86 | (0.82–0.9) |
| | 9 | 4368 | 10.1 | 1,478,427 | 11.3 | 0.85 | (0.81–0.88) | 0.83 | (0.79–0.87) |
| | 10 | 3982 | 9.2 | 1,532,279 | 11.8 | 0.74 | (0.71–0.78) | 0.79 | (0.75–0.83) |
| | Missing | 319 | 0.7 | 34,661 | 0.3 | | | | |
| Risk Groups | At risk (<65 years only) | 10,714 | 24.8 | 2,651,631 | 20.3 | 1.83 | (1.78–1.87) | 1.58 | (1.54–1.63) |
| | CEV[b] | 11,751 | 27.2 | 1,428,227 | 11.0 | 3.01 | (2.95–3.08) | 1.84 | (1.79–1.89) |
| | Severely Immunosuppressed | 2707 | 6.3 | 238,653 | 1.8 | 3.55 | (3.41–3.69) | 2.05 | (1.96–2.13) |
| | HSCW[c] | 1754 | 4.1 | 240,413 | 1.8 | 2.23 | (2.12–2.34) | 2.67 | (2.54–2.8) |

[a]IMD: Indices of multiple deprivation (1 (most deprived) to 10 (least deprived)).
[b]CEV: clinical extremely vulnerable.
[c]HSCW: health and social care worker.

**Table 2 | Description of data sources used to estimate vaccine effectiveness**

| Data source | Description | Covariates |
|---|---|---|
| Pillar 1 and 2 testing data | SARS-CoV-2 PCR testing in England is undertaken by hospital and public health laboratories (Pillar 1), as well as by community testing (Pillar 2). Pillar 1 testing data is available from UKHSA labs and NHS hospitals for those with a clinical need, and health and care workers. Pillar 2 community testing is available to anyone with symptoms consistent with COVID-19 (high temperature, new continuous cough, or loss or change in sense of smell or taste), anyone who is a contact with a confirmed case, care home staff and residents, and to those who have self-tested as positive using a lateral flow test (LFT). SGTF information to classify variants is available from some Pillar 2 tests. | Date of birth (age), period (week of the test), variant status (where it was determined by SGTF or period), previous positivity |
| NIMS vaccination record | The National Immunisation Management System (NIMS) contains demographic information on the whole population of England who is registered with a general practice physician in England and is used to record all COVID-19 vaccinations. | Vaccination status, index of multiple deprivations, ethnicity, geographic region, date of birth, health and social care worker status, clinical risk group status, clinically extremely vulnerable, severely immunosuppressed, care home status |
| Genomics data | Sequencing and genotyping are undertaken through a network of laboratories, including the Wellcome Sanger Institute. Whole-genome sequences are assigned to UKHSA definitions of variants based on mutations. Data were provided by the Genomics Cell at UKHSA. | Variant status (sequencing and genomics) |
| Secondary Care Hospital Admission Data (SUS) | SUS is the national electronic database of hospital admissions that provides timely updates of ICD-10 codes for completed hospital stays for all NHS hospitals in England. Up to 24 ICD-10 diagnosis fields can be completed in SUS for each admission with the first diagnosis field indicating the primary reason for admission. Length of stay was calculated as the date of discharge–date of admission. Where multiple admissions were linked to the same sample date the first admission after the sample date was retained and episode length was calculated by summing the stay length for each admission. Data were restricted to those with ARI in the first diagnosis field and where the length of stay was at least two days. | |
| Emergency Care Hospital Admission Data (ECDS) | Emergency Care hospital admissions include hospital admissions through emergency departments but not elective admissions. Admissions due to an injury were excluded. Admissions were identified where the Emergency Care Destination code was either discharged to a ward, intensive care unit, coronary care unit, high dependency unit, or where there was a date on which the decision to admit the patient was made. Admissions with an acute respiratory illness (ARI) SNOMED coded as the reason for attending emergency care were flagged. | |

Cases (those testing positive) and controls (those testing negative) were defined from the testing data. Data were linked to the NIMS to ascertain vaccination status, to the genomics line list to ascertain variant information, and to the SUS and ECDS hospitalisation records.

**Table 3 | Effectiveness of the ChAdOx1-S and BNT162b2 booster vaccines against symptomatic disease following infection with the Omicron variant for adults aged 40 years and older in England**

| Age (years) | Dose | Booster manufacturer | Interval (days) | Controls | Cases | OR[a] | VE[b] (95% CI) |
|---|---|---|---|---|---|---|---|
| 40–64 | Unvaccinated | | | 27,361 | 51,265 | Baseline | Baseline |
| | Dose 2[c] | n/a | 175+ | 85,175 | 89,230 | 0.92 (0.9–0.94) | 8 (6–9.9) |
| | Booster | Any[d] | 0–1 | 11,879 | 7715 | 0.8 (0.77–0.83) | 20.3 (17.2–23.3) |
| | | Any[d] | 2–6 | 27,430 | 21,422 | 0.74 (0.72–0.76) | 25.8 (23.7–27.8) |
| | | BNT162b2 | 7–13 | 28,809 | 17,658 | 0.42 (0.41–0.43) | 58.2 (57.0–59.4) |
| | | BNT162b2 | 14–34 | 86,719 | 66,406 | 0.36 (0.35–0.37) | 63.8 (63.0–64.5) |
| | | BNT162b2 | 35–69 | 87,592 | 90,787 | 0.43 (0.42–0.44) | 57.3 (56.4–58.2) |
| | | BNT162b2 | 70–104 | 22,504 | 29,379 | 0.54 (0.52–0.55) | 46.4 (45.0–47.8) |
| | | BNT162b2 | 105+ | 2758 | 4278 | 0.69 (0.66–0.73) | 30.6 (26.8–34.3) |
| | | ChAdOx1-S | 7–13 | 70 | 40 | 0.39 (0.25–0.59) | 61.2 (40.9–74.6) |
| | | ChAdOx1-S | 14–34 | 193 | 159 | 0.48 (0.38–0.61) | 51.7 (38.9–61.8) |
| | | ChAdOx1-S | 35–69 | 216 | 215 | 0.47 (0.38–0.57) | 53.0 (42.6–61.6) |
| | | ChAdOx1-S | 70–104 | 69 | 97 | 0.59 (0.43–0.81) | 40.8 (18.6–56.9) |
| | | ChAdOx1-S | 105+ | 10 | 14 | 0.63 (0.27–1.44) | 37.2 (−44.1 to 72.6) |
| 65+ | Unvaccinated | | | 1701 | 2361 | Baseline | Baseline |
| | Dose 2[c] | n/a | 175+ | 4466 | 3053 | 0.81 (0.73–0.88) | 19.5 (11.7–26.6) |
| | Booster | Any[d] | 0–1 | 428 | 110 | 0.65 (0.5–0.85) | 34.6 (14.8–49.8) |
| | | Any[d] | 2–6 | 1140 | 370 | 0.71 (0.61–0.84) | 28.6 (16.0–39.3) |
| | | BNT162b2 | 7–13 | 1883 | 433 | 0.42 (0.36–0.48) | 58.1 (51.6–63.8) |
| | | BNT162b2 | 14–34 | 14,311 | 3010 | 0.31 (0.29–0.34) | 68.5 (65.7–71.2) |
| | | BNT162b2 | 35–69 | 36,300 | 25,240 | 0.46 (0.42–0.49) | 54.1 (50.5–57.5) |
| | | BNT162b2 | 70–104 | 14,210 | 18,317 | 0.6 (0.55-0.65) | 40.1 (35.2–44.5) |
| | | BNT162b2 | 105+ | 1970 | 2789 | 0.77 (0.7–0.85) | 23.1 (15.1–30.5) |
| | | ChAdOx1-S | 7–13 | 23 | 8 | 0.34 (0.14–0.83) | 66.1 (16.6–86.3) |
| | | ChAdOx1-S | 14–34 | 53 | 32 | 0.48 (0.3–0.79) | 51.6 (20.8–70.4) |
| | | ChAdOx1-S | 35–69 | 88 | 81 | 0.56 (0.4–0.78) | 44.5 (22.4–60.2) |
| | | ChAdOx1-S | 70–104 | 16 | 40 | 1.27 (0.7–2.32) | −27.2 (−131.6 to 30.1) |
| | | ChAdOx1-S | 105+ | 3 | 5 | 0.98 (0.23–4.28) | N too low |

[a]Odds ratio.
[b]Vaccine effectiveness.
[c]ChAdOx1-S primary course.
[d]ChAdOx1-S or BNT162b2.

**Table 4 | Effectiveness of the ChAdOx1-S and BNT162b2 booster vaccines against hospitalisation as defined by linkage to the secondary care (SUS) data following infection with Delta or Omicron variants in adults aged 65 years and older in England**

| Dose | Booster manufacturer | Interval (days) | Controls | Cases | OR[a] | VE[b] (95% CI) |
|---|---|---|---|---|---|---|
| *Delta* | | | | | | |
| Unvaccinated | | | 1327 | 2000 | Baseline | Baseline |
| Dose 2[c] | | 175+ | 7000 | 3248 | 0.27 (0.24–0.3) | 73.4 (70.4–76.2) |
| Booster | Any[d] | 0–6 | 1123 | 243 | 0.13 (0.11–0.16) | 86.8 (83.7–89.2) |
| | BNT162b2 | 7+ | 7884 | 355 | 0.06 (0.05–0.07) | 93.9 (92.8– 94.9) |
| | ChAdOx1-S | 7+ | 43 | 4 | 0.19 (0.04–0.84) | 80.9 (15.6– 95.7) |
| *Omicron* | | | | | | |
| Unvaccinated | | | 517 | 603 | Baseline | Baseline |
| Dose 2[c] | | 175+ | 1596 | 430 | 0.39 (0.3–0.5) | 61.0 (49.8–69.7) |
| Booster | Any[d] | 0–6 | 361 | 27 | 0.24 (0.13-0.45) | 75.5 (54.9– 86.7) |
| | BNT162b2 | 7+ | 6495 | 953 | 0.09 (0.07–0.11) | 90.9 (88.7– 92.7) |
| | ChAdOx1-S | 7+ | 52 | 18 | 0.18 (0.09–0.36) | 82.3 (64.2– 91.3) |

[a]Odds ratio.
[b]Vaccine effectiveness.
[c]ChAdOx1-S primary course.
[d]ChAdOx1-S or BNT162b2.

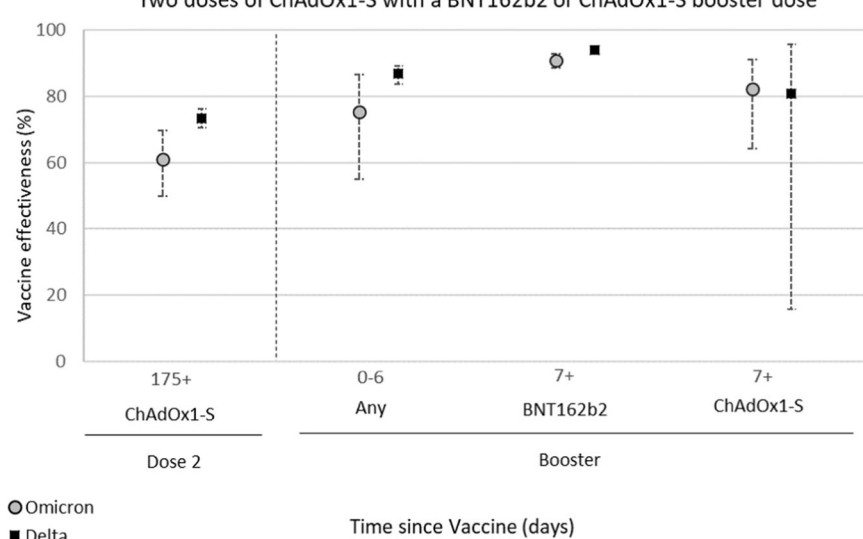

**Fig. 1 | Vaccine effectiveness estimates with 95% confidence intervals of the ChAdOx1-S and BNT162b2 booster vaccines against hospitalisation as defined by linkage to the secondary care (SUS) data following infection with Delta (black squares) or Omicron (grey circles) variants in adults aged 65 years and older in England.** The ChAdOx1-S booster vaccine enhanced protection against hospitalisation following infection with both Delta and Omicron from seven days after the vaccine was received. Protection was lower than that observed following a BNT162b2 booster but still very high at 82.3% (64.2–91.3%) against the Omicron variant a week after receiving a ChAdOx1-S booster, as compared to the unvaccinated. Source data are provided as a Source Data file.

The analyses assessing model robustness gave post-booster estimates all within ±2% of those reported except for VE against hospitalisation with the Omicron variant for those aged 40–64 who received a CHAdOx-S1 booster. Here, VE was 6% higher and required additional inclusion of ethnicity and healthcare worker status to be within 2%.

## Discussion

Here, we find reassuring real-world evidence that booster vaccination with ChAdOx1-S following a primary ChAdOx1-S course provides increased protection against mild and severe diseases with Delta and Omicron infection. Protection against symptomatic disease with Omicron was broadly similar to the protection seen with BNT162b2, with substantial waning by 10 or more weeks. Protection against the symptomatic disease was similar for both vaccines among 40–64-year olds but lower with ChAdOx1-S in older adults. Vaccine effectiveness against hospitalisation was high after a booster dose of ChAdOx1-S at over 80%, though lower than that observed following a BNT162b2 booster.

Boosting with ChAdOx1-S is not recommended as standard in the UK COVID-19 vaccination programme and there were notable differences between individuals who received ChAdOx1-S and BNT162b2 booster doses. Those who received three doses of ChAdOx1-S were more likely to be in risk groups; this is in line with the national recommendation that only those who were clinically contraindicated to get an mRNA booster should get a ChAdOx1-S booster[20]. From previous studies in the UK, we know that VE is lower in clinically vulnerable populations[13]. Although this risk status was adjusted for in the VE estimates, there is likely to be residual confounding which could have lowered the estimates of the ChAdOx1-S booster. Direct comparison of the protection conferred by the ChAdOx1-S as compared to BNT162b2 vaccination is therefore very challenging using data from the UK. Furthermore, very few people received a ChAdOx1-S booster as compared to a BNT162b2 booster, and the confidence intervals around the estimates for this group were large. Despite the risk of residual confounding, estimates remained similar with minimal confounding adjustment and in matched analyses.

Nonetheless, boosting with ChAdOx1-S improved protection against symptomatic disease following Omicron infection from the waning observed at 25 weeks post the second dose in both younger (40–64 years) and older (65 years and older) adults. Our findings are consistent with data from COV-BOOST in the UK and from Thailand which found both homologous and heterologous boosting with ChAdOx1-S enhanced immunity as measured by anti-spike IgG and neutralisation assays[6,15]. COV-BOOST found that boosting with BNT162b2 resulted in higher titres of anti-spike IgG, a greater cellular response as well as increased neutralising capacity as compared to ChAdOx1-S[15]. Early real-world data from Brazil also indicate that the ChAdOx1-S booster confers a high level of protection against severe disease[22]. Together, these data support the use of the ChAdOx1-S booster vaccines to prevent severe disease, including against the Omicron variant. Significant waning was observed in the period after booster vaccination, indicating that further boosters will be required if the aim of a vaccination programme is to prevent infection or mild disease. The only analysis where there was a big discrepancy in VE between the ChAdOx1-S boosted group and the BNT162b2 boosted group was symptomatic disease with Delta in those aged 65 years and older. It is unclear why this result differs from the other analyses, however, the Delta analyses were based on small numbers who had received a ChAdOx1-S booster.

The Omicron variant has previously been demonstrated to evade immunity from previous infection and vaccination[1], however reassuringly the ChAdOx1-S booster still provided high levels of protection against severe disease with this variant. Previously we have found that using broader definitions of hospitalisation has given lower VE estimates, reflecting outcome misclassification where cases are likely coincidentally positive whilst in hospital, without this being the reason for admission[23]. This is likely to be the reason for the lower estimates using the ECDS data. Protection against severe disease defined more specifically as requiring at least 2 days stay in hospital and a respiratory code in the primary diagnostic field using the SUS data demonstrated that protection from severe disease was very high amongst those who received the ChAdOx1-S booster. Numbers were too small to look at more severe and specific endpoints such as those who required oxygen, ventilation, or intensive care. In previous studies, using these endpoints yielded higher vaccine effectiveness estimates[23].

This study supports the consideration of ChAdOx1-S as a booster for protection against severe disease with COVID-19 in settings that have not yet offered booster doses and suggests that those who received ChAdOx1-S as a booster in the UK do not require re-vaccination ahead of others. Any consideration of the use of ChAdOx1-S should take into account this effectiveness data alongside existing vaccine safety assessments and local epidemiology. Comparison with other vaccines is challenging due to the different populations that have received each vaccine in the UK and head-to-head trials are likely to be required for a robust comparison.

## Methods

### Study design

To estimate vaccine effectiveness against symptomatic disease and hospitalisation, a test-negative case-control design was used. The odds of vaccination in symptomatic PCR-positive cases were compared to the odds of vaccination in symptomatic individuals aged 40 years and older who tested negative for SARS-CoV-2 in England.

### Data sources

Full details of the data sources used are in Table 2, and full details of the data linkage methodology are illustrated in Supplementary Fig. 1.

### COVID-19 testing data

SARS-CoV-2 PCR testing in England is undertaken by hospital and public health laboratories (Pillar 1), as well as by community testing (Pillar 2) (Table 1).

Data on all positive PCR and lateral flow tests (LFTs), and on negative Pillar 2 PCR tests from symptomatic individuals with a test date after 25 November 2020 were extracted up to 17 February 2022. Any negative tests were taken within 7 days of a previous negative test, and any negative tests where the symptom onset date was within the 10 days or a previous symptoms onset date for a negative test were dropped as these likely represent the same episode. Negative tests taken within 21 days of a subsequent positive test were also excluded as chances are high that these are false negatives. Positive and negative tests within 90 days of a previous positive test were also excluded; however, where participants had later positive tests within 14 days of a positive then preference was given to PCR tests and symptomatic tests. Data were restricted to persons who had reported symptoms and gave a symptom onset date within the 10 days before testing to account for reduced PCR sensitivity beyond this period in an infection event.

### Vaccination data

To estimate the odds of an individual receiving a ChAdOx1-S booster following a ChAdOx1-S primary course in the general population, data on all individuals aged 18 years and older who had received a ChAdOx1-S primary course and either a ChAdOx1-S or BNT162b2 booster dose were extracted from the NIMS vaccination record (Table 1) on 14 March 2022.

To estimate vaccine effectiveness, testing data were linked to NIMS on 21 February 2021 using combinations of the unique individual National Health Service (NHS) number, date of birth, surname, first name, and postcode using deterministic linkage—97.6% of eligible tests could be linked to the NIMS (Table 1). NIMS was accessed for dates of vaccination and manufacturer, sex, date of birth, ethnicity, and residential address. Addresses were used to determine the index of multiple deprivation quintiles and were also linked to Care Quality Commission registered care homes using the unique property reference number. Data on geography (NHS region), risk group status, clinically extremely vulnerable status, and health/social care worker were also extracted from the NIMS. Clinical risk groups included a range of chronic conditions as described in the Green Book[20], whereas the CEV group included persons who were considered to be at the highest risk for severe COVID-19, including those with immunosuppressed conditions and those with severe respiratory disease. The CEV flag is a record of vulnerable patients thought to be at high risk of complications from COVID-19. The NHS number is used to extract data from the GP electronic health record (EHR), Hospital Episode Statistics (HES), and the QCOVID risk stratification assessment. The severely immunosuppressed flag is also provided by NHS Digital and includes those with specific immunosuppression conditions. The severely immunosuppressed were offered additional doses as part of the primary vaccination schedule which the CEV was not.

Booster doses were identified as a third dose given at least 84 days after a second dose and administered after 13 September 2021. Individuals with four or more doses of vaccine, heterologous primary schedule, or fewer than 19 days between their first and second dose were excluded.

### Identification of Delta and Omicron Variants and assignment to cases

Sequencing of PCR-positive samples is undertaken through a network of laboratories, including the Wellcome Sanger Institute. Whole-genome sequences are assigned to UKHSA definitions of variants based on mutations[24,25]. S-gene target status on PCR testing is an alternative approach for identifying variants because Delta has been associated with a positive S-gene target status on PCR testing with the Taqpath assay while Omicron BA.1 has been associated with S-gene target failure (SGTF). Omicron BA.2 has also been associated with positive S-gene target status but can be distinguished from Delta by sequencing and period. Cases were defined as Delta or Omicron (BA.1 and BA.2) based on whole genome sequencing, genotyping, SGTF, or period, with sequencing taking priority, followed by genotyping, SGTF status, and period. Where subsequent positive tests within 14 days included sequencing, genotyping, or S-gene target failure information, this information was used to classify the variant. The Delta analysis was restricted from 13 September 2021 to 9 January 2022. The Omicron analysis was restricted from 29 November 2021 to 17 February 2022.

### Hospital admission data

Hospital inpatient admissions for a range of acute respiratory illnesses were identified from the Secondary Uses Service (SUS)[26] (Table 1) and were linked to the testing data on 22 February 2022 using NHS number and date of birth. For the Pillar 2 samples, admissions with an ICD-10 acute respiratory illness (ARI) discharge diagnosis in any diagnosis field (Supplementary Table 1) was identified where the sample was taken 14 days before and up to 2 days after the day of admission. For the Pillar 1 samples, admissions with an ICD-10 coded ARI discharge diagnosis in any diagnosis field were identified where the sample was taken 1 day before and up to 2 days after the admission. Data were restricted to those with ARI in the first diagnosis field and where the length of stay was at least two days. The data was restricted to tests up to 2 February 2022 to account for delays in the SUS data recording.

As a secondary analysis, Emergency Care hospital admissions from the Emergency Care Dataset (ECDS)[27] (Table 1), which includes hospital admissions through emergency departments but not elective admissions, were linked to the testing data using NHS number and date of birth on 22 February 2022 to identify admissions within 14 days of the sample date. The data were restricted to tests up to 6 February 2022 to allow sufficient follow-up time.

Since the Omicron variant which causes milder disease became dominant, an increasing proportion of cases in hospitals are likely to have COVID-19 as an incidental finding rather than as the primary reason for admission[23]. Previously, we have found that using the ECDS data gives estimates that are likely more reflective of VE against infection[23]. As such, we regard the SUS analysis as the primary analysis but also include the ECDS analysis to allow comparison to previous studies. Supplementary Table 2 describes the data sources used for each outcome investigated in further detail.

Control selection For analyses involving hospitalised controls any negative tests that led to a hospitalisation within 21 days of a previous hospital negative test were excluded. A maximum of one negative test per person within each of the following approximate 3-month periods was selected at random: 26 April to 1 August 2021, 2 August 2021–21 November 2021, and 22 November 2021–2 February 2022. For analyses involving all Pillar 2 symptomatic controls, the same was done within this control group.

## Statistical analysis

Logistic regression was used to estimate differences in demographic and clinical characteristics of individuals who received a ChAdOx1-S booster as compared to those who received a BNT162b2 booster. The logistic regression model adjusted for age (in 5-year bands, then everyone age 90 years or older), sex, index of multiple deprivations (decile), ethnic group, geographic region (NHS region), health and social care worker status, clinical risk group status (only available for those aged 64 years and younger, CEV status and severely immunosuppressed status[20,28].

To estimate vaccine effectiveness, logistic regression was used with the PCR test result as the dependent variable and cases being those testing positive (stratified in separate analyses as either Omicron or Delta and as either age 40–64 years or 65 years and older) and controls being those testing negative. Under 40-year-olds were not included, given that ChAdOx1-S was not recommended in the general population in this age group. Vaccination status was included as an independent variable and effectiveness was defined as 1−odds of vaccination in cases/odds of vaccination in controls. Vaccine effectiveness was adjusted in logistic regression models for age (ages 40 through to 89 in 5-year bands, then everyone age 90 years or older), sex, index of multiple deprivations (quintile), ethnic group, geographic region (NHS region), period (week of the test), health and social care worker status, clinical risk group status, clinically extremely vulnerable, severely immunosuppressed and previously testing positive. These factors were all considered potential confounders so were included in all models (Table 1 for the description of covariates).

To evaluate the robustness of estimates, models were assessed with minimal adjustment (period and age and pillar) and also using a matched analysis for the hospital endpoints with matching on age, week of the sample, risk group, and CEV and adjustment for other covariates.

## Software

Data linkage and manipulation were performed using Microsoft SQL Server Management Studio 18 and statistical analysis was performed using Stata 15.

## Ethics Committee approval

UKHSA Research Ethics and Governance Group Statement: Surveillance of COVID-19 testing and vaccination is undertaken under Regulation 3 of The Health Service (Control of Patient Information) Regulations 2002 to collect confidential patient information (http://www.legislation.gov.uk/uksi/2002/1438/regulation/3/made) under Sections 3(i) (a) to (c), 3(i)(d) (i) and (ii) and 3[3]. The study protocol was subject to an internal review by the PHE Research Ethics and Governance Group and was found to be fully compliant with all regulatory requirements. As no regulatory issues were identified, and ethical review is not a requirement for this type of work, it was decided that a full ethical review would not be necessary.

All necessary patient/participant consent has been obtained and the appropriate institutional forms have been archived.

## Reporting summary

Further information on research design is available in the Nature Portfolio Reporting Summary linked to this article.

## Data availability

This work is carried out under Regulation 3 of The Health Service (Control of Patient Information) (Secretary of State for Health, 2002))[3] using patient identification information without individual patient consent. Data cannot be made publicly available for ethical and legal reasons, i.e. public availability would compromise patient confidentiality as data tables list single counts of individuals rather than aggregated data. Databases used in this study include the National Immunisation Management System (NIMS), the Unified Sample Database, and the Emergency Care Dataset (ECDS). Aggregated source data (the minimum dataset) are provided in this paper. The raw vaccine effectiveness data are protected and are not available due to data privacy laws. Source data are provided with this paper.

## Code availability

Available upon request.

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

## Author contributions

F.C.M.K. and J.L.B. wrote the manuscript. J.L.B., N.A. and M.R. conceptualised the study. F.C.M.K., J.S., R.S. curated the data. R.S., N.A., F.C.M.K. conducted the demographic analysis. N.A. conducted the VE analysis. All co-authors reviewed the manuscript.

## Competing interests

The authors declare no competing interests.
