## [Peer Review File · Nature Communications]

Effectiveness of ChAdOx1-S COVID-19 Booster Vaccination against the Omicron and Delta variants in EnglandReviewer #1 (Remarks to the Author):

In this manuscript the authors outline the VE of the ChAdOx Booster vaccine against the omicron and delta variants in the UK. The data are interesting and encouraging but additional caveats and details should be added to the manuscript.

In the Abstract the authors need to add the numbers of subjects who received the ChAdOx booster, particularly since they mention that these were a unique population with small numbers. Also the comparison numbers of those who received the mRNA vaccine should also be added. Are the sample sizes adequate to conclude that receipt of the ChAdOx vaccine booster does not need revaccination?

In the Introduction on line 25, please add the number of ChAdOx booster doses administered if available. Also given the rare adverse thrombotic events associated with the ChAdOx vaccine and the restrictions on its use in the UK, should it be recommended for use in LMIC?

In the Methods, on line 68-69, abbreviation for LFTs should be spelled out and more detail regarding the Pillar 2 PCR tests should be provided. The exclusion criteria for the test negative and test positive samples appear appropriate as do the links with the vaccination data. Line 101 needs a year added to the date, 29 November "2021". Lines 103-105 need clarity as to how Pillar 1 and Pillar 2 differ. The authors should consider adding a table outlining the data sources and definitions of illness in the main body of the paper and not only in the supplementary appendix. Line 128 CEV needs a definition and how it differs from severely immunosuppressed.

In the Results section Line 147-148 we are first given the numbers of subjects included. This should be added to the Abstract. Those who received the ChAdOx booster were different than those who received mRNA as outlined in lines 142-152. These factors are likely associated with a lesser serologic response for the immunosuppressed and CEV.

The authors need to describe why only those > 40 years were included in the VE estimates. Were there no individuals < 40 years who received the booster? Please clarify. Were there any subjects vaccinated with mRNA in the primary series and boosted with ChAdOx? Were these data analyzed?

The sample sizes available for the studies of VE for hospitalization had fewer subjects in the positive cases. Were these numbers adequate to assess these differences? Lines 175-178 are confusing without a better understanding of the Pillar 1 and Pillar 2 tests.

In the Discussion the authors caution that the confidence intervals around these VE estimates are large reflecting the limited number of subjects enrolled in some arms of the study. Also the reason that these numbers are small is the restriction on the use of the vaccine in the UK for safety reasons. Do the authors think that these same safety concerns should restrict its use in LMIC as well?

Does it appear that waning is faster for the ChAdOx than mRNA?

Reviewer #2 (Remarks to the Author):

In this manuscript, Kirsebom et al assessed the effectiveness of both ChAdOx-1 S and BNT162b2 booster vaccination after a primary regimen with ChAdOx-1 S among adults in England. This is an impressive effort of database linking and mining, although I have some comments the authors should address.

Major comments

- With the increasing circulation of Omicron sublineages it becomes more important to specify. The authors did distinguish between BA.1 and BA.2 breakthrough infections according to their methods, but do not report vaccine effectiveness separately. They

should clarify.

- Both in the abstract and introduction the authors are a bit vague on how widespread the use of ChAdOx-1 S as a booster vaccination is. Should be clearer. For example: line 2: "despite the potential widespread global use" or line 24-25: 1.6 billion boosters were given, but how many of these were ChAdOx1-S?

- The authors should be careful when using the word "immunity" to describe immune responses. For example, in line 5, line 220 or line 229.

- I find it weird that the authors almost 'downplay' their BNT162b2 data. It is suddenly mentioned in line 13 of the abstract. It would be nice if the BNT162b2 data forms an integral part of the manuscript, not as an 'add-on'. Especially because in the discussion, line 214-215, the authors speculate that their BNT162b2 is potentially stronger than the ChAdOx-1 S data.

- Similarly, if I read the results properly, effectiveness against symptomatic disease is only assessed for the Omicron variant, whereas effectiveness against hospitalization is done for Delta and Omicron. Why is that?

- The last sentence of the abstract is not supported by data from the authors. ChAdOx-1 S performs worse than BNT162b2 according to their own data, and longevity was not properly studied.

- Line 116: The Omicron variant is not milder, the disease course caused by the omicron variant is milder. Rephrase.

Minor comments

- The abstract is not a good reflection of the paper: Delta is not mentioned at all.

- Line 3-4 of the abstract are unclear: "highly effective" against what? Disease? Hospitalization? Transmission? Which variant?

- Check abbreviations. CEV? LFT?

- COVID-19 infection is incorrect. Change into SARS-CoV-2 infection (line 23)

- Line 37-38: What does 'highly immunogenic' mean? Effective? Induces high levels of antibodies? The authors should be clearer.

- I think the graph in figure 1 is potentially useful for visualizing all the VE data that the authors collected, but in its current form is not really easy to read. Can the authors improve?

- Line 33: "waning against symptomatic disease.." should be waning of protection against symptomatic disease.

- Line 201: "disease with Omicron was broadly similar to that seen with BNT162b2". Similar to what?

Reviewer #3 (Remarks to the Author):

This is a very well written manuscript conveying a very useful and clear message on the clinical effectiveness of ChadOx1 vs mRNA booster vaccination. The data used for these analyses are unique in many ways, and the information obtained is of great relevance for public health authorities internationally

I have a few comments that I would like to see addressed or clarified:

MAJOR

1.- given the clear guidelines on boosting preferentially with mRNA vaccines, it is highly likely that the exposure to either vaccine was informative, leading to confounding by indication. if unresolved, this will lead to bias. Although the authors adjusted for a number of confounders, there is a risk that residual confounding remained at least partially responsible for the findings. ideally, one would like to see additional measures

and/or diagnostics and/or sensitivity analyses used to minimise or at least measure the impact of confounding, e.g. negative control exposures, assessment of additional confounders like medicine/s use or comorbidity, use of matching (instead of multivariable adjustment), probabilistic analyses to look at the potential impact of unobserved confounding, etc

2.- the recommendation to use mRNA vaccines for boosters is based on safety concerns with ChAdOx1. it would be of great importance to see if the exposure groups differed in terms of vte or coagulopathy history, and ideal if some measure of safety eg rates of vte or ate could be obtained from this unique dataset

MINOR

1.- the authors have great expertise in the analysis of these linked datasets, but readers will feel lost at times trying to figure out what the study population is, and where the utilised variables (including covariates used for adjustment) were obtained. it would be most useful if a population flowchart could be provided, and a table (maybe in an appendix) could be prepared, listing each covariate vs the data source where they were obtained

Reviewer #1 (Remarks to the Author):

In this manuscript the authors outline the VE of the ChAdOx Booster vaccine against the omicron and delta variants in the UK. The data are interesting and encouraging but additional caveats and details should be added to the manuscript.

In the Abstract the authors need to add the numbers of subjects who received the ChAdOx booster, particularly since they mention that these were a unique population with small numbers. Also the comparison numbers of those who received the mRNA vaccine should also be added. Are the sample sizes adequate to conclude that receipt of the ChAdOx vaccine booster does not need revaccination?

The numbers of individuals who received ChAdOx1-S and BNT162b2 booster vaccines have now been added to the Abstract. The 95% confidence interval demonstrate 83% protection against hospitalisation with a bottom end of the confidence interval at 64%. We believe this is sufficiently high not to require revaccination.

In the Introduction on line 25, please add the number of ChAdOx booster doses administered if available. Also given the rare adverse thrombotic events associated with the ChAdOx vaccine and the restrictions on its use in the UK, should it be recommended for use in LMIC?

The number of ChAdOx booster doses administered globally is not available since many countries do not report vaccine coverage by dose or manufacturer.

We have not made recommendations specific for LMICs. Our recommendation that ChAdOx is suitable as a booster is based on our analysis of vaccine effectiveness and it would be up to individual National Immunisation Technical Advisory Groups to assess the risks vs benefits depending on local burden. While ChAdOx has been rarely associated with VITTS, similar rare but serious adverse events (e.g. Myocarditis) have been associated with mRNA vaccines and each requires risk vs benefit assessments. We have changed the wording to suggest "consideration" of ChAdOx as a booster to avoid any assumption that use of the vaccine as a booster should be based on our effectiveness data alone.

In the Methods, on line 68-69, abbreviation for LFTs should be spelled out and more detail regarding the Pillar 2 PCR tests should be provided. The exclusion criteria for the test negative and test positive samples appear appropriate as do the links with the vaccination data. Line 101 needs a year added to the date, 29 November "2021".

LFT has now been spelled out the first time it is used in the text (line 197). Further detail on Pillar 1 and Pillar 2 testing has been moved from the Supplementary Appendix and added under COVID-19 Testing Data in Methods/Data Sources and in Table 2. 29 November has been amended to 29 November 2021.

Lines 103-105 need clarity as to how Pillar 1 and Pillar 2 differ. The authors should consider adding a table outlining the data sources and definitions of illness in the main body of the paper and not only in the supplementary appendix.

We have now added further clarification on the differences between Pillar 1 and Pillar 2 in the text and in Table 2. Table 2 describes the data sources in more detail, and we have removed this from the Appendix. We have also added Supplementary Table 2 to describe the outcomes (definitions of illness) investigated.

Line 128 CEV needs a definition and how it differs from severely immunosuppressed.

The following clarification has now been added in the methods, “The CEV flag is a record of vulnerable patients thought to be at high risk of complications from COVID-19. The NHS number is used to extract data from the GP electronic health record (EHR), Hospital Episode Statistics (HES), and the QCOVID risk stratification assessment. The severely immunosuppressed flag is also provided by NHS Digital and includes those with specific immunosuppression conditions. The severely immunosuppressed were offered additional doses as part of the primary vaccination schedule which the CEV were not.”

In the Results section Line 147-148 we are first given the numbers of subjects included. This should be added to the Abstract. Those who received the ChAdOx booster were different than those who received mRNA as outlined in lines 142-152. These factors are likely associated with a lesser serologic response for the immunosuppressed and CEV.

The number of individuals who received ChAdOx1-S and BNT162b2 booster vaccines has now been added to the Abstract.

The authors need to describe why only those > 40 years were included in the VE estimates. Were there no individuals < 40 years who received the booster? Please clarify. Were there any subjects vaccinated with mRNA in the primary series and boosted with ChAdOx? Were these data analyzed?

There were 520 individuals <40 years who received a ChAdOx primary course followed by a ChAdOx booster who reported symptoms and had a PCR test, but these individuals were excluded as individuals aged <40 years were not recommended to receive ChAdOx as a primary course in the UK*. This is highlighted in the methods on lines 285-286. There were 114 subjects vaccinated with mRNA primary series and boosted with ChAdOx in the datasets to analyse VE against symptomatic disease for the Omicron variant. These were excluded as there were so few.

*JCVI statement on use of the AstraZeneca COVID-19 vaccine: 7 April 2021.

<https://www.gov.uk/government/publications/use-of-the-astrazeneca-covid-19-vaccine-jcvi-statement/jcvi-statement-on-use-of-the-astrazeneca-covid-19-vaccine-7-april-2021>

The sample sizes available for the studies of VE for hospitalization had fewer subjects in the positive cases. Were these numbers adequate to assess these differences?

The 95% confidence intervals indicate precision which we think is sufficient, even if they are quite wide.

Lines 175-178 are confusing without a better understanding of the Pillar 1 and Pillar 2 tests.

Further information on Pillar 1 and Pillar 2 testing has now been added to the Methods and in Table 2.

In the Discussion the authors caution that the confidence intervals around these VE estimates are large reflecting the limited number of subjects enrolled in some arms of the study. Also the reason that these numbers are small is the restriction on the use of the vaccine in the UK for safety reasons. Do the authors think that these same safety concerns should restrict its use in LMIC as well?

The reason that ChAdOx1-S was not used as a booster in the UK for over 40 year olds was not due to safety concerns, rather a consideration at the time that mRNA vaccines were more effective. We have reviewed the discussion and couldn't find a suggestion that the vaccine as a booster had been

restricted due to safety concerns. All of the COVID-19 vaccines have safety concerns and effectiveness data needs to be considered alongside any rare adverse events and local epidemiology. We have added a sentence to the conclusion (lines 179-181) to make this clear.

Does it appear that waning is faster for the ChAdOx than mRNA?

It is not clearly different when looking at the 95% confidence intervals but precision is low at the longer intervals.

Reviewer #2 (Remarks to the Author):

In this manuscript, Kirsebom et al assessed the effectiveness of both ChAdOx-1 S and BNT162b2 booster vaccination after a primary regimen with ChAdOx-1 S among adults in England. This is an impressive effort of database linking and mining, although I have some comments the authors should address.

Major comments

- With the increasing circulation of Omicron sublineages it becomes more important to specify. The authors did distinguish between BA.1 and BA.2 breakthrough infections according to their methods, but do not report vaccine effectiveness separately. They should clarify.

Previously we have not found a difference in VE between BA.1 and BA.2*. Therefore, we combined BA.1 and BA.2 cases for the Omicron analyses. This has now been clarified on lines 76-77.

* Kirsebom, F. C. M., et al. (2022). "COVID-19 vaccine effectiveness against the omicron (BA.2) variant in England." The Lancet Infectious Diseases **22**(7): 931-933.

- Both in the abstract and introduction the authors are a bit vague on how widespread the use of ChAdOx-1 S as a booster vaccination is. Should be clearer. For example: line 2: "despite the potential widespread global use" or line 24-25: 1.6 billion boosters were given, but how many of these were ChAdOx1-S?

The number of ChAdOx booster doses administered globally is hard to ascertain as global coverage data by dose and manufacturer isn't easily accessible.

Line 1 in the abstract has now been amended, and it has been clarified that the total number of ChAdOx booster doses administered globally is unclear in the Introduction.

- The authors should be careful when using the word "immunity" to describe immune responses. For example, in line 5, line 220 or line 229.

Line 5, line 220, and line 229 have been amended for clarity.

- I find it weird that the authors almost 'downplay' their BNT162b2 data. It is suddenly mentioned in line 13 of the abstract. It would be nice if the BNT162b2 data forms an integral part of the manuscript, not as an 'add-on'. Especially because in the discussion, line 214-215, the authors speculate that their BNT162b2 is potentially stronger than the ChAdOx-1 S data.

It was not our intention to downplay the BNT162b2 data, but our intention to make the ChAdOx data the focus of the manuscript given how much has been published already (by us and others) on VE for BNT162b2. We agree that the abstract doesn't adequately describe the BNT162b2 data which we do present and this is now mentioned sooner.

- Similarly, if I read the results properly, effectiveness against symptomatic disease is only assessed for the Omicron variant, whereas effectiveness against hospitalization is done for Delta and Omicron. Why is that?

We prioritised showing VE against both the Omicron and Delta variants for the hospitalisation outcome as we think these are the most important data. We now included data on VE against symptomatic disease with the Delta variant in Supplementary Table 4. Here we use just one category for time since the booster dose (7 days post booster) since we did not have enough cases to estimate VE against symptomatic disease using the granular intervals post the booster dose as we show in Table 3 for the Omicron variant. In the older adults we find VE for the ChAdOx1-S booster is lower than VE for the BNT162b2 booster. We do not see this in the VE estimates against hospitalisation with the Delta variant where the point estimate is lower for the ChAdOx1-S booster but 95% confidence intervals overlap. The confidence intervals are wide for the VE estimates for the ChAdOx1-S booster against symptomatic disease with Delta (56.3% (30.0 to 72.7%)) so this finding may be due to small numbers.

- The last sentence of the abstract is not supported by data from the authors. ChAdOx-1 S performs worse than BNT162b2 according to their own data, and longevity was not properly studied.

In our study both vaccines are highly effective against severe disease and the difference in their effectiveness is non-significant – the 95% confidence intervals overlapped. We have changed the final sentence of the Abstract to say that this study supports the “consideration” of ChAdOx1-S as a booster to protect against severe disease. Each individual National Immunization Technical Advisory Group would need to take into account effectiveness data alongside a wide range of other factors including local epidemiology, vaccine availability, vaccine safety and cost. We have also added this to the Discussion for clarity.

- Line 116: The Omicron variant is not milder, the disease course caused by the omicron variant is milder. Rephrase.

This has now been re-phrased.

Minor comments

- The abstract is not a good reflection of the paper: Delta is not mentioned at all.

The abstract has been re-worded, also to fit with the 150 word-limit. VE for hospitalisation with Delta is now mentioned on lines 8-10.

- Line 3-4 of the abstract are unclear: “highly effective” against what? Disease? Hospitalization? Transmission? Which variant?

The abstract has been amended and this sentence was removed to keep within the word limit.

- Check abbreviations. CEV? LFT?

CEV and LFT are now spelt out the first time they are used in the text.

- COVID-19 infection is incorrect. Change into SARS-CoV-2 infection (line 23)

This has now been amended.

- Line 37-38: What does ‘highly immunogenic’ mean? Effective? Induces high levels of antibodies? The authors should be clearer.

This has now been amended.

- I think the graph in figure 1 is potentially useful for visualizing all the VE data that the authors collected, but in its current form is not really easy to read. Can the authors improve?

We are happy with this figure as it is but if the reviewer has any specific suggestions on changes they would like us to make to improve clarity, please let us know.

- Line 33: “waning against symptomatic disease..” should be waning of protection against symptomatic disease.

This has now been amended.

- Line 201: “disease with Omicron was broadly similar to that seen with BNT162b2”. Similar to what?

This has now been amended to ‘disease with Omicron was broadly similar to *the protection seen with BNT162b2*’

Reviewer #3 (Remarks to the Author):

This is a very well written manuscript conveying a very useful and clear message on the clinical effectiveness of ChadOx1 vs mRNA booster vaccination. The data used for these analyses are unique in many ways, and the information obtained is of great relevance for public health authorities internationally

I have a few comments that I would like to see addressed or clarified:

MAJOR

1.- given the clear guidelines on boosting preferentially with mRNA vaccines, it is highly likely that the exposure to either vaccine was informative, leading to confounding by indication. if unresolved, this will lead to bias. Although the authors adjusted for a number of confounders, there is a risk that residual confounding remained at least partially responsible for the findings. ideally, one would like to see additional measures and/or diagnostics and/or sensitivity analyses used to minimise or at least measure the impact of confounding, e.g. negative control exposures, assessment of additional confounders like medicine/s use or comorbidity, use of matching (instead of multivariable adjustment), probabilistic analyses to look at the potential impact of unobserved confounding, etc

We agree that it is possible some residual confounding may remain. We did not have access to additional variables for adjustment (although in a separate unpublished piece of work looking at data from earlier in the vaccine roll-out we have looked at this based on a questionnaire sent to cases and controls and found results are similar with additional confounder adjustment). We also do not have access to an obvious negative control exposure to use. What we have done is assessed the effectiveness in models with minimal adjustment (just age and period and Pillar 1 or 2) as this gives an indication how much estimates do depend on the other covariates. We have also run matched analyses for hospital VE with matching on age, period and risk group and with adjustment on other covariates. We did not do this on the symptomatic infection VE as matched analyses would not converge due to the size of the group matching. These analyses gave estimates of VE post booster all within 2% of those found using the full models with one exception where inclusion of ethnicity and health care worker status was also required for estimates to be within 2% of the reported VE. This does provide some reassurance that estimates are fairly robust.

We also agree a simulation model to assess the potential impact of unobserved confounding for test negative studies would be interesting and this is something we are interested in doing in the future.

This has now been added to the manuscript in the Results (line 123-126), Discussion (line 147-148) and Methods (line 294-296).

2.- the recommendation to use mRNA vaccines for boosters is based on safety concerns with ChAdOx1. it would be of great importance to see if the exposure groups differed in terms of vte or coagulopathy history, and ideal if some measure of safety eg rates of vte or ate could be obtained from this unique dataset

We agree this is of interest in the study of vaccine safety but would not be part of this paper.

MINOR

1.- the authors have great expertise in the analysis of these linked datasets, but readers will feel lost at times trying to figure out what the study population is, and where the utilised variables (including covariates used for adjustment) were obtained. it would be most useful if a population flowchart could be provided, and a table (maybe in an appendix) could be prepared, listing each covariate vs the data source where they were obtained

We have now included Table 2 – a full description of data sources, and which covariates were derived from each. We have also included Supplementary Table 2 to describe the outcomes investigated (and the data sources used to investigate each outcome) more clearly. Supplementary Figure 1 has been added to illustrate the linkage methodology.

Reviewer #1 (Remarks to the Author):

The authors have been responsive to the comments of the reviewers and the manuscript. I have no additional comments to make.

Reviewer #2 (Remarks to the Author):

The authors have sufficiently addressed my major and minor concerns In this revised version of the manuscript, I have no further comments.

Reviewer #3 (Remarks to the Author):

Thank you for responding to my comments. I hope that your changes have improved the manuscript.